# Climate Performance, Environmental Toxins and Nutrient Density of the Underutilized Norwegian Orange-Footed Sea Cucumber (*Cucumaria frondosa*)

**DOI:** 10.3390/foods12010114

**Published:** 2022-12-26

**Authors:** Andreas Langdal, Karl-Erik Eilertsen, Marian Kjellevold, Eldbjørg S. Heimstad, Ida-Johanne Jensen, Edel O. Elvevoll

**Affiliations:** 1The Norwegian College of Fishery Science, Faculty of Biosciences, Fisheries and Economics, UiT—The Arctic University of Norway, N-9037 Tromsø, Norway; 2Institute of Marine Research, P.O. Box 1870, Nordnes, NO-5817 Bergen, Norway; 3NILU—Norwegian Institute for Air Research, The Fram Centre, N-9296 Tromsø, Norway; 4Department of Biotechnology and Food Science, Norwegian University of Science and Technology, NTNU, N-7491 Trondheim, Norway

**Keywords:** low trophic, daily recommended nutrient intake, *bêche-de-mer*, carbon dioxide equivalents, life cycle assessment (LCA), orange-footed sea cucumber, environmental pollutants, climate impact, arsenic

## Abstract

Low trophic species are often mentioned as additional food sources to achieve broader and more sustainable utilisation of the ocean. The aim of this study was to map the food potential of Norwegian orange-footed sea cucumber (*Cucumaria frondosa*). *C. frondosa* contained 7% protein, 1% lipids with a high proportion of polyunsaturated fatty acids, and a variety of micronutrients. The nutrient density scores (NDS) of *C. frondosa* were above average compared towards daily recommended intakes (DRI) for men and women (age 31–60) but below when capped at 100% of DRI. The concentrations of persistent organic pollutants and trace elements were in general low, except for inorganic arsenic (iAs) (0.73 mg per kg) which exceeded the limits deemed safe by food authorities. However, the small number of samples analysed for iAs lowers the ability to draw a firm conclusion. The carbon footprint from a value chain with a dredge fishery, processing in Norway and retail in Asia was assessed to 8 kg carbon dioxide equivalent (CO_2_eq.) per kg *C. frondosa*, the fishery causing 90%. Although, *C. frondosa* has some nutritional benefits, the carbon footprint or possible content of iAs may restrict the consumption.

## 1. Introduction

The international ambition to reduce global carbon emissions was reaffirmed at the UN Climate Change Conference in Glasgow (COP26) in 2019. The report “Global warming of 1.5 °C” from The Intergovernmental Panel on Climate Change (IPCC) states that keeping the peak global warming to 1.5 °C would grant clear benefits to reduce climate-related risks [1]. Currently, food production emits between 26% to 33% of total human societal greenhouse gas (GHG) emissions [2,3], with wild capture fisheries emitting 4% of the global food systems production emissions [4].

GHG emissions should not be seen as a solitary challenge. The human population has grown rapidly in recent centuries and has more than tripled since the 1950s [5]. In 2015 we faced for the first time in a decade an increase in global undernourishment. Entailing that 9% of the global population were severely food insecure (unable to fulfil their energy needs), and 25% were moderately to severely food insecure (struggling or worrying about access to a healthy balanced diet) [6]. Alongside, there is a growth in poor quality diets and, consequently, an increase in lifestyle-dependent diseases [7], such as ischaemic heart disease and stroke, which now are the leading causes of death globally [8]. A common concern has been the rise of energy-rich but nutrient-poor diets [9]. Further expansion of sustainable and nutritious food alternatives is essential. As of 2017, 6.2% of the fish stocks assessed by the Food and Agriculture Organization of the United Nations (FAO) are deemed underfished whereas 34.2% are overfished [10]. In Norway, one underfished species is the *C. frondosa* [11,12]. *C. frondosa* has been harvested for decades in North America and Iceland [13,14]. However, there is no distinct fishery documented in Norway.

Several parts of the biology and ecological traits of *C. frondosa* are known, with stages like larval development, growth, locomotion, and reproduction already well documented [15]. The nutritional properties of Canadian *C. frondosa* have also been studied, highlighting that it contains mostly protein, polysaccharides, and ash (on a dry weight basis), and the fatty acid profile is dominated by the omega-3 long-chain polyunsaturated fatty acid, eicosapentaenoic acid (EPA, 20:5 n-3) [16,17]. Song et al. [17] also found the heavy metals mercury (Hg), cadmium (Cd), arsenic (As) and lead (Pb) to be within safety limits [17]. It should also be noted that the current Norwegian stock estimate of *C. frondosa* was measured to be sustainable with low to no risk of extinction in 2021 [18]. Fishing is currently prohibited by Norwegian law, however, permission can be granted by the Norwegian Directorate of Fisheries for harvest inside the Norwegian sea baseline [19]. To the authors knowledge, no studies have previously analysed the contents of environmental pollutants in the edible part of *C. frondosa*. Likewise, carbon emissions through the production of *bêche-de-mer* are fairly unexplored.

The aim of this research was to assess the food potential of Norwegian *C. frondosa* for human consumption. This assessment was done by analysing the composition of micro- and macronutrients; a nutrient grading to highlight how the nutrient composition compares with other animal protein sources; an analysis of relevant pollutants (persistent organic pollutants (POPs) and trace elements); and the CO_2_eq. emissions of a potential fishery and its related value chain. The most prevalent food product of sea cucumber is gutted and dried body wall, “*bêche-de-mer*” [15,20], thus all analysis was performed on gutted Norwegian *C. frondosa*. The nutrient grading were ranked according to a nutrient density score (NDS) from Hallström et al. [21]; the pollutants was compared with maximum recommended consumption levels; and the emissions were estimated by following the ISO standard 14044: Environmental management—Life cycle assessment—Requirements and guidelines [22] to create a life cycle assessment (LCA). The nutrient density and emissions of *C. frondosa* were then compared with Atlantic herring (*Clupea harengus*), Atlantic salmon (*Salmo salar*) and Norwegian beef (Norwegian Red (dual purpose cattle)) as traditional food resources. By laying such a foundation, it can be evaluated how beneficiary the utilisation of this resource could be.

## 2. Materials and Methods

### 2.1. Sampling and Pre-Treatment

The *C. frondosa* (specimens (*n*) = 61) was collected by divers at 12–15 m depth at the mouth of Balsfjord, Northern Norway (69°28′29.62″ N, 18°56′5.77″ E) in April 2022 by the R/V Hyas. The *C. frondosa* were stored in seawater for a few hours until delivered at UiT—The Arctic University of Norway and transferred to tanks with continuous saltwater supply. The specimens were stored alive in these tanks for 3–4 days at 4 °C before internal organs, mouth, anus, and aquapharyngeal bulb were removed (Figure 1), and the body-wall was frozen at −80 °C. Before analysis, the body-walls were cut into smaller pieces and grounded with an HR1364/00 hand mixer (Philips, Amsterdam, The Netherlands). The samples were re-frozen to −80 °C and freeze-dried at about −80 °C with a VirTis Genesis 35 EL (VirTis SP Scientific, Warminster, PA, USA) continuously for 48 h. The samples were then grounded to a powder with an A 11 basic Analytical mill (IKA, Königswinter, Germany) and were the basis for all analysis except water and ash content. 

### 2.2. Proximate Analyses

#### 2.2.1. Water and Ash Content

Water content was measured according to Association of Analytical Chemists (AOAC) 950.46B [23] and attained by drying 10 g fresh body-wall (four duplicates, three replicates) at 105 °C until weight stabilized in a heratherm oven (ThermoFisher Scientific, Waltham, MA, USA). Ash content was measured according to AOAC 938.08 [23] and attained by combusting the dried samples at 540 for 16 h in a muffle furnace (Nabertherm, Lilienthal, Germany). The water content was calculated with formula 1 while ash content was calculated with formula 2. All results are presented as percentage of wet weight body-wall.
(1)$$\frac{g\;\mathrm{water}\;}{100\;g\;\mathrm{sample}\;}\;=\frac{g\;\mathrm{sample}\;\mathrm{before}\;\mathrm{drying}\;-\;g\;\mathrm{sample}\;\mathrm{after}\;\mathrm{drying}}{g\;\mathrm{sample}\;\mathrm{before}\;\mathrm{drying}}\times 100\%$$
(2)$$\frac{g\;\mathrm{ash}}{100\;g\;\mathrm{sample}\;}=\frac{g\;\mathrm{sample}\;\mathrm{after}\;\mathrm{drying}\;\mathrm{and}\;\mathrm{combustion}}{g\;\mathrm{sample}\;\mathrm{before}\;\mathrm{drying}\;\mathrm{and}\;\mathrm{combustion}}\times 100\%$$

#### 2.2.2. Total Fat and Fatty Acid Composition

The fat content was extracted with a modified version of Folch et al. [24] following the procedures of Dalheim et al. [25]. In short, 0.15 g freeze-dried samples (four duplicates, three replicates) were mixed with 0.1 mL (5.05 mg per ml) heptadecanoic acid (Supelco Analytical, Bellefonte, PA, USA) as internal standard, in 2:1 dichloromethane (DCM) (VWR chemicals, Leicestershire, UK): methanol (MeOH) (VWR chemicals, Leicestershire, UK), and 2.9 mL DCM:MeOH. The samples were shaken in a Multi Reax (Heidolph Instrument, Schwabach, Germany) at 600 rpm and added 3 mL 5% NaCl in MilliQ-water (Merck KGaA, Darmstad, Germany), shaken for 1 min by hand, and centrifuged at 4000× *g* with a multifuge 1 S-R (Heraeus, Hanau, Germany) for 5 min. The organic liquid at the bottom of the sample tube was thus extracted. The remaining material in the sample tube was reextracted two more times with an additional 2.5 mL DCM and centrifuged at 4000× *g* for 5 min each time. The liquid organic matter was then flushed dry with N_2_ gas by a Sample Concentrator SBCONC/1 (Stuart-equipment, Staffordshire, UK). Total fat content was calculated with the use of formula 3. Results are presented in percentage of wet weight body-wall.

The fatty acid composition was determined by methylation and gas chromatography according to a modified version of the method described by Christie and Han [26]. In short, lipid samples (four duplicates, three replicates) were diluted to 10 mg per ml in a solution of 2:1 DCM:MeOH before 100 μL were mixed with 900 μL DCM and 2 mL 2% H_2_SO_4_ (Honeywell—Fluka, Charlotte, NC, USA)) in MeOH and heated in a heating block (ThermoFisher Scientific, Waltham, MA, USA) at ~100 °C for one hour. Then, 3.5 mL heptane (VWR chemicals, Leicestershire, UK) and 3.5 mL 5% NaCl in water was added. The lipid layer was flushed dry with N_2_ gas by a Sample Concentrator SBCONC/1, dissolved in 100 μL heptane, and analysed by an Agilent 6890N Gas chromatograph equipped with a 7683B autoinjector and flame ionization detector (Agilent Technologies, Santa Clara, CA, USA). By comparing the samples fatty acids towards known standards PUFA no 1, PUFA no 3 (Sigma Chemicals Co, St. Louis, MO, USA) and GLC 502 (NuChec Prep. Inc., Elysian, MN, USA) specification was determined. Fatty acid content was calculated with the use of formula 4. Peak area fatty acid, including peak area heptadecanoic acid, was determined by the gas chromatograph software Agilent OpenLab CDS ChemStation Edition Rev. C.01.10 (287) (Agilent Technologies, Santa Clara, CA, USA). The fatty acid composition is presented in mg per 100 g body-wall *C. frondosa* and percentage of total fatty acids.
(3)$$\frac{g\;\mathrm{lipids}}{100\;g\;\mathrm{sample}\;}\;=\frac{g\;\mathrm{lipids}}{g\;\mathrm{sample}}\times 100\%$$
(4)$$\frac{g\;\mathrm{fatty}\;\mathrm{acid}}{100\;g\;\mathrm{sample}\;}=\frac{\mathrm{Peak}\;\mathrm{area}\;\mathrm{fatty}\;\mathrm{acid}}{\mathrm{Peak}\;\mathrm{area}\;\mathrm{heptadecanoic}\;\mathrm{acid}}*\;\frac{g\;\mathrm{heptadecanoic}\;\mathrm{acid}}{g\;\mathrm{sample}}\times 100\%$$

#### 2.2.3. Crude Protein Content and Amino Acids Composition

Amino acid content was analysed as done by Mæhre et al. [27] and hydrolysed according to Moore and Stein [28]. The total amount of protein was calculated as the sum of amino acid residues (the molecular weight of amino acid residue subtracted the molecular weight of water), as recommended by FAO [29]. To determine the amino acid composition, 40 mg freeze-dried material (four duplicates, two replicates) was mixed with 0.7 mL distilled deionized H_2_O from a Milli-Q: Millipore (Merck KGaA, Darmstad, Germany), 0.5 mL 20 mM internal standard L-Norleucin (Merck KGaA, Darmstad, Germany), and 1.2 mL 37% HCl. The samples were flushed with N_2_ gas for 15 s and heated in a Heratherm oven at 105–110 °C for 24 h. After heating, 1 mL was centrifuged on 14,000× *g* for 6 min, and 100 μL of the supernatant was dried with N_2_ gas through a Sample Concentrator SBCONC/1, and then dissolved with 1 mL 2.2 pH lithium citrate buffer. Samples were analysed with a Biochrom30+ Amino Acid Analyzer (Biochrom C, Cambringe, UK) with a lithium citrate-equilibrated column and post-column derivatization with ninhydrin (Ultra Ninhydin reagent kit, Biochrom, nr. 80-2118-30). Signals from the Biochrom 30+ were analysed with the Chromeleon Software and compared against physiological amino acids standards (Sigma, nr. A6407 and Sigma, nr. A6282). Amino acid content was calculated with the use of formula 5 with AA representing amino acid, and N-leu representing the internal standard. The amino acids and total amino acids (protein) are presented in mg per g body-wall *C. frondosa*.
(5)$$\frac{\mathrm{mg}\;\mathrm{AA}\;\mathrm{residue}\;}{g\;\mathrm{sample}}\;=\frac{\frac{\frac{\mathrm{nmol}\;\mathrm{AA}}{\mathrm{nmol}\;N-\mathrm{leu}}\times \mathrm{mL}\;N-\mathrm{leu}\;\times \;\mathrm{nmol}\;N-\mathrm{leu}}{g\;\mathrm{sample}\;}\times \mathrm{MW}\;\mathrm{AA}}{1000}\times \frac{\left(\mathrm{MW}\;\mathrm{AA}\;-\;\mathrm{MW}\;\mathrm{water}\right)\;}{\mathrm{MW}\;\mathrm{AA}}$$

#### 2.2.4. Trace Elements

The content of trace elements (silver (Ag), As, iAs, Cd, cobalt (Co), copper (Cu), iron (Fe), Hg, manganese (Mn), molybdenum (Mo), nickel (Ni), Pb, selenium (Se), vanadium (V), and zinc (Zn)) was determined based on NS-EN 16802:2016 and NS-EN 15111:2007 [30,31], by the Institute of Marine Research in Bergen, Norway. The methods are previously described by Wiech et al. [32]. In summary, samples were digested with 2 mL nitric acid (69% *w*/*w*) and an ultra-wave digestion system (UltraWAVE, Milestone, Sorisole, Italy). The tubes were capped and treated in an ultra-wave system with 130 mL Milli-Q (EMD Millipore Corporation, Billerica, MA, USA) water and 5 mL H_2_O_2_ and diluted to 25 mL with Milli-Q water. The ICP-MS was tuned and performed according to the manufacturer’s instructions. A tuning solution (1 ppb tuning solution B, Thermo Fisher, in 2% HNO_3_ and 0.5% HCl) was used before analyses. The concentrations of trace elements were determined by ICP-MS (iCapQ ICP-MS, Thermo Scientific, Waltham, MA, USA) equipped with an autosampler (FAST SC-4Q DX, Elemental Scientific, Omaha, NE, USA). Data were collected and processed using the Qtegra ICP-MS software (version 2.10, 2018, Thermo Scientific, Waltham, MA, USA). The dry weight-based limit of quantification (LOQ d.w.) was set to 0.005 mg/kg d.w. with a standard sample size (0.2 g). Results are presented in mg per kg body-wall *C. frondosa*.

Some trace elements (Cd, As, Pb, Hg and Se) were also analysed by NILU (Norwegian Institute for Air Research) using the methods previously described by Eilertsen et al. [33]. Samples were digested using an UltraClave (Milestone, Italy) microwave-assisted mineralization. Aliquots of 0.5–0.75 g of sample were placed in TFM tubes (GPE Scientific, Bedfordshire, England) added 5 mL concentrated supra pure nitric acid and 3 mL deionized water. Method blanks were produced using the same reagents but no sample in the TFM tubes. The samples, method blanks and CRMs (NIST 1566b Oyster tissue) were then treated to a four-step program maxing at 250 °C with hold time of 15 min. Post digestion, samples, method blanks and CRMs were transferred to polypropylene tubes and diluted to 50 mL with deionized water. All trace elements except mercury were analysed using an ICP-MS (Agilent 7700x, Agilent, Santa Clara, CA, USA). For As and Se, collision cell with helium gas (5.7 mL/min) was used. For Pb and Cd no gas mode was used. The elements were quantified by external calibration, using multielement mixtures made from stock solutions traceable to NIST. Indium was used as internal standard. To determinate mercury, aliquots of 25 mL extract were further diluted to 50 mL and topped with 1 mL BrCl before analyzed by cold vapour atomic fluorescence spectrophotometry (CV-AFS) (Tekran, Canada) according to method US-EPA-1631. The method blank samples were used to calculate limit of quantification (10 × standard deviation of method blanks). Method blank value was subtracted from all analytical results. Results will be presented in mg per kg body wall *C. frondosa.*

#### 2.2.5. Organic Pollutants

All organic contaminant analysis were performed by NILU-Norwegian Institute for Air Research. For the analysis of polychlorinated biphenyls (PCBs), all pentachloroben-zene (PeCB), hexachlorobenzene (HCB) and pesticide samples were prepared similarly. Briefly, 1.5 g freeze-dried *C. frondosa* were mixed and homogenized with a 20-fold amount of dry Na_2_SO_4_. Prior to extraction, the samples were added to a mixture of several different isotope labelled compounds for quantification purposes. The samples were extracted with organic solvents (cyclohexane/acetone, 1:1) and concentrated, followed by a sulphuric acid clean up and fractionation on a silica column to remove interferences before analysis. The compounds were quantified using GC/HRMS (EI) and/or GC-qToF (ECNI). Proper identification and quantification were confirmed based on correct retention time, correct isotope ratio, a signal/noise ratio > 3:1 and a correct recovery of internal standard, in addition to 15 accepted laboratory blanks for the method to monitor background levels and to detect possible contamination. The internal standards used were 13C-labelled standards of PCBs and pesticides: PeCB, HCB, PCB28, PCB52, PCB101, PCB105, PCB114, PCB118, PCB123, PCB 138, PCB153, PCB156, PCB157, PCB167, PCB180, PCB189, PCB209, a-HCH, b-HCH, g-HCH, p,p′-DDE, o,p′-DDD, p,p′-DDT. Results are presented in ng per g wet weight body-wall *C. frondosa*.

For dioxins and non-ortho PCBs (no-PCBs), extraction and clean-up were performed with a semi-automated three-column system as described in detail by Nash et al. [34]. In brief, 5 g of freeze-dried tissue was homogenized with anhydrous Na_2_SO_4_, spiked with internal standards (13C-labelled polychlorinated dibenzodioxins (PCDD), polychlorinated dibenzofurans (PCDF) and coplanar PCBs) and subjected to extraction and clean up through three columns prepared with (a) activated silica and potassium silica, (b) silica and (c) activated carbon with dichloromethane (DCM) and cyclohexane (1:1) followed by DCM. Finally, the PCDD, PCDF and coplanar PCBs were eluted from the column with activated carbon using toluene. The toluene ex-tracts were attributed to solvent exchange to hexane and further cleaned through consecutive sulphuric acid-coated silica column, followed by potassium hydroxide-coated silica column with hexane, followed by 1% DCM in hexane. 13C-labelled 1,2,3,4-TCDD recovery standard was added before analysis by HRGC-HRMSEI (an HP5890 GC coupled to a VG AutoSpec) by monitoring at *m/z* of the molecular ions. The separation of the congeners was carried out on a DB-5 ms (30 m, 0.25 mm, 11 µm film thickness) fused silica. Results are presented in pg TE per g wet weight body-wall *C. frondosa*.

For the analysis of perfluorinated alkylated substances (PFAS), the freeze-dried samples (0.4–0.9 g) were homogenized. Internal standards (shown in [35]) were added to the sample before it was extracted with methanol or acetonitrile using vortex and ultrasonication. After extraction the sample was concentrated followed by clean-up with emulsified carbon. The anionic PFASs were analyzed according to [35]. Briefly, the samples were analyzed by ultrahigh pressure liquid chromatography triple quadruple mass-spectrometry (UHPLC–MS/MS). Analyses were performed on a Thermo Scientific quaternary Accela 1250 pump, with a Waters Acquity UPLC HSS 3 T column (2.1 × 100 mm, 1.8 µm) coupled to a Thermo Scientific Vantage MS/MS (Vantage TSQ). Ionization was conducted in the negative electrospray ionization mode (ESI-). The QA/QC of the sample preparation and analysis was assured using mass labelled internal standards for (13C PFAS), where they were available. Quality of sample preparation and analysis for conventional PFASs was further assured with reference materials and laboratory blanks. Results are presented in ng per g wet weight body-wall *C. frondosa*.

For poly aromatic hydrocarbons (PAH) analysis, the samples were mixed with sodium sulphate, and transferred to a 1.5 cm ID glass column. Internal standards (Naphthalene, Acenaphthene, Anthracene, Pyrene, Benz(a)anthracene, Benzo(b)fluoranthenes, Benzo(ghi)perylene) were added followed by elution with a mixture of cyclohexane:ethyl acetate (1:1). The volume of the extracts was reduced to 0.5 mL and further cleaned by using a liquid-liquid extraction method followed by silica liquid column chromatography [36]. Determination of PAHs was carried out by a GC/LRMS on an Agilent7890B gas chromatograph coupled to an Agilent 5977A mass spectrometer in an electron impact (EI) mode. 1 μL was injected by an auto sampler on the split/splitless injection port in splitless mode with helium as a carrier gas (flow rate ~1 mL/min). Results are presented in ng per g wet weight body-wall *C. frondosa*.

#### 2.2.6. Micronutrient

The data for the micronutrient’s vitamin E, -B6, -B12, thiamin, riboflavin, niacin equivalents, and folate were analysed by the accredited laboratory at the Institute of Marine Research in Bergen, Norway. The methods are described by Reksten et al. [37]. The data for the micronutrients phosphorus, Fe, calcium (Ca), potassium (K), Cu, magnesium (Mg), Se, and Zn were attained from Song et al. [17].

#### 2.2.7. Nutrient Density Score Calculation

The nutrients were ranked in accordance with the NDS-C and NDS-G, from Hallström et al. [21]. Both scoring systems grade nutrients per 100 g of product with no weighing (increasing the value of specific nutrients). However, NDS-G applies capping (removing additional score from nutrient surplus higher than the daily recommended intake (DRI)), while NDS-C does not. The calculation was performed by summarising the nutrients divided by their DRI and subtracting the dis-qualifying nutrients divided by their maximum recommended intake (MRI), as shown in Equations (6) and (7). The qualifying nutrients, dis-qualifying nutrients, DRI and MRI are shown in Appendix A. The nutrients-score values for Atlantic herring, Atlantic salmon and Norwegian beef were calculated from data given by The Norwegian Food Safety Authority [38] and presented in Appendix B.
(6)$$\mathrm{NDS}-C:\;\sum_{i=1}^{x}\frac{Nutrient\;i}{DRI\;i}-\sum_{j=1}^{y}\frac{Nutrient\;j}{MRI\;j}$$
(7)$$\mathrm{NDS}-G:\;\sum_{i=1}^{x}(if\left(\frac{Nutrient\;i}{DRI\;i}\right)>1,\;1,\;\left(\frac{Nutrient\;i}{DRI\;i}\right))-\sum_{j=1}^{y}\frac{Nutrient\;j}{MRI\;j}$$

**Theorem** **1.**
*NDS is the nutrient density score, x is the number of qualitative nutrients, y is the number of dis-qualitative nutrients, i is the amount of nutrients, and j is the amount of dis-qualitative nutrients. DRI is the daily recommended intake of qualitative nutrients and MRI is the maximum recommended intake. All calculations are done per 100 g of uncooked products.*


### 2.3. Carbon Dioxide Equivalent Assessment

Since there are no commercially active *C. frondosa* fisheries or producers in Norway, the calculations have been performed with values adopted from the closest relatable production chains in Canadian and Icelandic *C. frondosa* fisheries/industries. Where this was not found to give representable data for the estimates, other relatable production chains were used. The emissions will be estimated from cradle (fishery) to retail. Post-retail emissions were not analysed due to their high variability and low data availability [2,39]. However, one exception was made with the addition of nutrient loss at rehydration at consumer level, as this was considered invariable, independent of post-retail treatment. Sea cucumbers are highly valued in The People’s Republic of China, Japan, and South Asia as an important seafood delicacy [17]. In this paper, the retail stage was chosen to be in The People’s Republic of China. A flowchart of the production chain is shown in Figure 2. The functional unit used was 1 kg body-wall *C. frondosa*. The emissions was estimated with carbon dioxide equivalents (CO_2_eq.) for a 100-year period of global warming potential (GWP), as given by IPCC [40]. Emissions related to by-products from the production chain used in other products were allocated according to mass, equally to the main product. In contrast, emissions related to by-products not utilised in other products were allocated to the main product.

#### 2.3.1. Carbon Dioxide Assessment Model within Fishing

A novel experimental sea cucumber fishery in Norway has chosen to use a modified Icelandic dredge [41]. An estimate was therefore made by expecting a catch of 540 kg per hour (±271) of *C. frondosa* per day—the average catch per unit effort for Icelandic fishers in 2018 [13] and time at sea of 7.3 h as found in a sea cucumber fishery logbook [42]. Fuel usage varies between 500–3000 L depending on ship size [43]. However, common ship sizes vary from 13–30 m [15,20]. Average fuel efficiency was thus estimated to be 0.39 L fuel per kg catch [43,44,45,46,47]. In recent years, the simultaneous use of two dredges has become common, and energy usage was therefore calculated with two dredges at 180% fuel efficiency [13]. For fuel type, it was expected that marine gas/diesel oil would be used, as it is the most predominant fuel type in Norwegian demersal fishing [48].

Sea cucumbers autolyze when stressed or taken out of seawater, so iced seawater is recommended as a transport medium to avoid skin necrosis and ensure quality [49]. Thus, chilled seawater was expected to be used. The fuel used for cooling is expected to be negligible [50]. Traditional fishery cooling agents have notably high carbon footprints [39]. As of 2017, ammonia (potential negative GWP, but noted with low confidence by the IPCC) and CO_2_ (GWP of 1 kg CO_2_eq.) are the most common cooling agents in Norwegian fisheries [40,50,51]. However, several fishing vessels have introduced “drop in” mediums instead of traditional cooling agents [40]. For “drop in” mediums, the closest relatable fishery was found to be demersal trawlers, where Winther et al. [48] estimated an emission of 0.007 kg hydrofluorocarbons (HFC) per ton landed. Without knowing which mediums were used, hydrochlorofluorocarbon-22 (HCFC-22/R22) (GWP of 1760 kg CO_2_eq. per kg), a traditional cooling medium was used as a benchmark [40]. Even though the footprint of the agents used as “drop in” refrigerants could have more than twice the GWP [51].

#### 2.3.2. Carbon Dioxide Assessment Model within Processing

Handling of *C. frondosa* post catch may differ. Canadian governmental research recommends evisceration and immediate cooling in seawater post-capture [14]. On the other side, other literature points out that the *C. frondosa* should be kept alive until delivery on land for evisceration [15]. Gutting has traditionally been done with knife and spoon to open and remove the guts [52,53,54]. This trend was still found in Canadian *C. frondosa* processing plants [55,56]. Internal organs are usually discarded as waste, while the body-wall is kept as the main product [57]. There are some discrepancies regarding the gutting yield of *C. frondosa*. Some sources estimate a 50% yield [14,57], while others estimate about 26% [58]. The sea cucumbers from this study were found to have a 28% yield. This yield was thus used for the LCA.

For processing, cooking before drying has been a common treatment of *C. frondosa* [14,15,20,52,57]. Cooking was determined according to Sonesson et al. [59]. For drying calculations, two calculations were used after discussions with a representative from Algetun AS, a Norwegian producer of dried red sea cucumber (*Parastichopus tremulus*). Drying efficiency from the industrial producer Munter’s at 1.25 kWh per kg of water and drying efficiency from Algetun AS at 0.58 kWh per kg of water [60]. A 10% moisture content was expected in the finished dried product [15]. The values from Munter’s were used as baseline and values from Algetun were used as minimum estimate. As other literature has pointed out [48], only 14% of Norwegian electricity purchased has guarantees of origin. Norwegian electricity will be highly influenced by the European electricity grid. A footprint of 0.25 kg CO_2_eq. per kWh was thus expected [61].

#### 2.3.3. Carbon Dioxide Assessment Model for Product Storage

To estimate emissions during storage, a generic storage scenario for fish was given in Winther et al. [39]. This results in energy use for storage at 0.438 kJ per kg per day [39], though 6.4 × 10^−8^ kg refrigerant hydrofluorocarbons-134a (HFC-134a) per kg fish per day from Winther et al. [48] was added. HFC-134a has a GWP of 1300 CO_2_eq. per kg [40].

#### 2.3.4. Carbon Dioxide Assessment Model within Transport

Transport was expected to go to Shanghai as China’s most active port [62]. However, the product might have an intermediate stop in Rotterdam, Netherlands, as in similar export estimates [39]. According to Winther et al. [48] transport emissions were assumed to be 57 g CO_2_eq. per ton product per km to Rotterdam when transported on inland waterways, with a barge with reefer and kept frozen. For trans-oceanic sea transport with cooling to Shanghai, 22 g CO_2_eq. per ton product per km was assumed. In contrast, the Network for Transport Measures [63] (NTM), an environmental performance calculator for traffic and transport estimates 30 g CO_2_eq. per ton product per km between Tromsø–Rotterdam (regional containership at 2000 ton carry capacity) and 22.8 g CO_2_eq. per ton product per km between Rotterdam–Shanghai (oceanic containership at 25,000 ton carry capacity). Since NTM estimates are based upon baseline data [63], data from Winther et al. [48] was used as an estimate, whereas NTM was used as a minimum value.

The *C. frondosa* was expected to be transported in bags of 250 g dried sea cucumber/1000 cm^3^ (1 L) by comparing commercial packages [55,56,64] to available packages. They were then expected to be packed in 2 kg cardboard boxes. The size of the cardboard boxes were estimated to 0.0654 m^3^ [48]. Lastly, to ensure 1 kg of edible product, a 5% loss of nutrients was assumed at rehydration, as nutrients may leach out of the product during rehydration [65].

## 3. Results

### 3.1. Proximate Composition

The weight of whole and gutted *C. frondosa* was 640 ± 172 and 180 ± 46 g (*n* = 61), respectively, entailing a 28% body-wall yield ratio. Lipid, protein, ash, and water content in gutted *C. frondosa* represented 1%, 7%, 3%, and 83%, respectively (Table 1), in total 94% of the nutrients.

#### 3.1.1. Fatty Acid Composition

The saturated fatty acids (SFA), monounsaturated fatty acids (MUFA), and polyunsaturated fatty acids (PUFA) represented 21%, 16% and 48%, respectively, of the total fatty acid amount (Table 2). The primary fatty acids identified were eicosapentaenoic acid (EPA) (C20:5 n-3) at 29%, and stearic acid (C18:0) and stearidonic acid (C18:4 n-3) both at 8% of the total fatty acid content. The total content of omega-3 fatty acids was approximately five times higher than that of omega-6 fatty acids. The total amount of unidentified fatty acids was 36 mg per 100 g *C. frondosa*.

#### 3.1.2. Amino Acid Composition

The protein content was 7 g per 100 g of gutted *C. frondosa*, consisting of 2 g of essential amino acids. The primary amino acids were glutamic acid at 16%, arginine at 15% and glycine at 12%. The ratio of essential amino acids to total amino acids was at 28% (Table 3).

### 3.2. Environmental Pollutants

The most prevalent trace metals were Zn, Fe, and As at 8, 4 and 2 mg per kg wet weight *C. frondosa*. Nearly 50% of the As is present as iAs (Table 4). Trace minerals analysed by the Institute of Marine Research and NILU are within one standard deviation of each other’s results. Polychlorinated dibenzo-para-dioxins/polychlorinated dibenzofurans (PCDD/PCDF) were the major toxic equivalent (TE) factor (82%) out of sum dioxin-like PCB (dl-PCB) and PCDD/PCDF (Table 5). All 23 per- and polyfluoroalkyl substances (PFAS) compounds were below limit of detection (<0.008 to <0.03 ng/g sample) (Table 6).

### 3.3. Carbon Dioxide Equivalent Assessment

The estimated CO_2_eq. emissions for a modeled *C. frondosa* production chain in Norway to The People’s Republic of China is shown in Figure 3. The total emissions were estimated to almost 8 kg CO_2_eq. per kg body-wall. The production stage with the highest emission estimate is fishery with almost 7 kg CO_2_eq. per kg body-wall (high estimate: 10 kg; low estimate: 5 kg). The second highest stage is processing at 0.6 kg CO_2_eq. per kg body-wall (low estimate: 0.5 kg). The transport emission was estimated to be close to 0 kg (109 g) CO_2_eq. per kg body-wall (low estimate: 100 g). Storage had the lowest emission estimate at less than 0.0 kg (2 g) CO_2_eq. per kg body-wall.

### 3.4. Nutrient Density

The qualitative and dis-qualitative nutrients for NDS-C and G according to Hallström et al. [21] in 100 g of body-wall *C. frondosa* (Table 7). Fibre, retinol eq., vitamin D, ascorbic acid, iodine, and sodium were not accessible.

The NDS-C and NDS-G compared with the total emissions per 100 g body-wall are shown in Figure 4 for *C. frondosa*, herring, salmon and beef. *C. frondosa* has seven times the emissions of herring, 120% of the salmon, and 28% of beef per 100 g product. NDS-C and NDS-G have a far lower spread in beef and salmon compared to herring and *C. frondosa*. The NDS is generally similar between genders except for NDS-C for *C. frondosa* where the score is 11% higher for women than men.

## 4. Discussion

It is estimated that by 2050 we need to ensure 47% more food than produced in 2020 [67]. Increased utilisation of marine resources is recommended as one solution to feed the growing global population [68]. An increase in consumption of low trophic marine resources have been recommended in both Norwegian and European context [69,70]. Sea cucumber products have seen an ever-growing demand, leading several wild sea cucumber stocks to depletion. Strategies for farming or novel fisheries are thus essential to fulfill future demands [17]. *C. frondosa* is an underutilised resource in Norway. It is deemed a sustainable stock [18], thus of great interest. However, *C. frondosa* fisheries show fluctuating landings from region to region and year to year in Iceland and Canada [13,71]. Precautionary principles to avoid overfishing are thus needed if such fisheries are to be developed in Norwegian waters.

### 4.1. Nutrient Content

The high water-, low lipid-, and mediocre protein contents makes the *C. frondosa* a lean food source with 1.1% lipids, 6.7%protein, 3.1% ash, and 82.8% water, respectively. Our findings deviate slightly compared with other research. Zhong et al. [16] found 0.5, 8.3, 2.97, and 87.4 g per 100 g body-wall wet-weight Canadian *C. frondosa* of lipids, protein, ash, and water, respectively. Our results on water and ash content are thus similar, thought, our finding of 1.1% lipids is twice that of Zhong et al. [16]. The protein found in Norwegian *C. frondosa* was slightly lower than the findings by Zhong et al. [16], reporting a 24% higher content. *C. frondosa* contains all essential amino acids. However, the levels of essential amino acids are only about one quarter of each essential amino acid compared to traditional animal protein sources [72]. The proportion of EPA is noteworthy (70 mg/100 g product) and similar to Atlantic cod (90 mg/100 g) and Northern prawn (100 mg/100 g product) [38]. Further, *C. frondosa* contains a wide range of micronutrients, and 100 g of body-wall ensures the DRI of vitamin B12 and vitamin E; and more than 40% of the DRI for Se. Its nutritional contribution may thus benefit consumers in need of dietary energy reduction. The sum of lipid, protein, ash, and water was 94%, and slightly lower than the FAO standard [73]. *C. frondosa* have been found to contain around 11% polysaccharides on a dry weight basis [17]. Thus when including the polysaccharides, our findings are within an acceptable range according to FAO standards [73].

### 4.2. Nutrient Score

Herring is documented to be among the most nutritious animal protein sources, whereas salmon is close to average, and beef is below average [21]. The NDS of *C. frondosa* is higher than salmon with NDS-C and similar to beef with NDS-G. The utilisation of *C. frondosa* can thus contribute to nutrient diversity equally to other common food items. The NDS is relatively similar between genders, with a slightly higher NDS for women, particularly for the *C. frondosa* with NDS-C. As many nutritional recommendations correlate with body weight, the generally lower weight of women [74] results in a generally higher score for these nutrients. This also means that products will score higher in regions with lower average body weight, such as Asia [75]. A deviation is in folate and iron where the NDS scores higher for men, as these nutrients are recommended in higher amounts for women of reproductive age [74]. However, the main difference between genders for *C. frondosa* with NDS-C is connected to the high vitamin E content (48 times the content in herring), which results in a gender score discrepancy. As NDS-G reduces nutrients score above the DRI, this discrepancy is reduced.

The benefit of utilising NDS-C compared to NDS-G is its representation of the total nutritional content of the food source, while NDS-G artificially caps it at specific values. However, NDS-G gives a better representation of the nutrient content for human consumption (water-soluble nutrients will be wasted in over-consumption, and lipid-soluble nutrients might be detrimental to human health at high consumption [76]). In this case, *C. frondosa* contains almost three times the DRI of vitamin E and four times the DRI of vitamin B12. Vitamin B12 is water-soluble, and a majority of it is thus wasted. Over-consumption of Vitamin E can on the other hand, cause side-effects. However, the toxicity of natural vitamin E is low, and no adverse effects have been described from food sources [74].

Nutrient data for sodium, retinol (vitamin A), vitamin D, ascorbic acid (vitamin C) and iodine, were not accessible to fulfill the NDS methodology [21]. However, calculating using only the same nutrients for herring and *C. frondosa*, the herring scored 11% lower with NDS-C and 18% lower with NDS-G (independent of gender). The *C. frondosa* scores 15% lower than herring with NDS-C for men when comparing the same nutrients (25% lower with all nutrients) and 12% lower with the same nutrients for women (22% lower with all nutrients). For NDS-G, *C. frondosa* scores 35% lower with the same nutrients for men (48% lower with all nutrients) and 37% lower with the same nutrients for women (49% lower with all nutrients).

The NDS calculations for Figure 4 were based on the minerals found in Song et al. [17] instead of the values from Table 4. This was done to ensure that the minerals required for the calculations came from the same source. This influenced the NDS score slightly, as the values in Table 4 showed three times the Cu, four times the Se, and 50% more Zn in comparison. Using the values in Table 4, the *C. frondosa* would score roughly 20% better on both NDS-C and NDS-G, bringing the score of NDS-C closer to that of herring.

### 4.3. Food Safety

There are no specified maximum limits (ML) of the content of Hg, Cd, Pb and As in Echinodermata. However, some guidelines are given by FAO/WHO [77] and the European Commission (EC) on maximum levels for contaminants in marine food items. The maximum content of Hg in unspecified fish muscles and fishery products, including molluscs and crustaceans, is set at 0.5 mg/kg wet weight (ww). The European maximum limit of Cd is set at 0.05 mg/kg ww in the muscle of unspecified fish but as high as 1.0 mg/kg ww for bivalve molluscs [78]. The European maximum limit of Pb is set at 0.3 mg/kg ww in the muscle of most fish species but as high as 1.5 mg/kg ww for molluscs [78]. The levels of Hg, Cd, and Pb in *C. frondosa* (Table 4) are thus lower than the limits for comparable marine species. There is, at present, no guideline or ML for total As in seafood in Europe, but commonly consumed seafood, assessed as suitable for human consumption, such as molluscs (4.0 ± 3.6 mg/kg ww), pelagic fish (6.5 ± 7.2 mg/kg ww) and demersal fish (5.1 ± 5.4 mg/kg ww) have higher concentrations of As compared to *C. frondosa* (Table 4) [79]. Worldwide regulations for As and iAs in food were reviewed in detail by Petursdottir et al. [80], and for The People’s Republic of China (the most prevalent market of sea cucumbers), a maximum level of 0.5 mg/kg iAs in “Aquatic animals and products (excluding fish and fish products)” was set in 2017 [81]. As in seafood is generally present in the form of organic compounds but some organisms may contain high levels of iAs [82]. A recent paper by Gajdosechova et al. [83] presented total As and noted worrying levels of iAs species in *C. frondosa* with 70% of the recovered As, as iAs. Body-wall from Canadian fisheries at the Atlantic coast, Nova Scotia and Newfoundland and Labrador contained 1.4 and 1.7 mg iAs per kg dry weight. Using the water content (82.7%) from our analyses this would compare to 0.2 and 0.3 mg per kg wet weight. In 2009, The European Food Safety Authority (EFSA) revised the provisional tolerable weekly intake (PTWI) of iAs (0.015 mg/kg body weight) from all sources [84]. Using the lowest consumption level reported without harm (0.3 μg/kg body weight) [84], an adult of 60 kg should not consume more than 25 g per day of *C. frondosa* (using our results for iAs (Table 4)) if *C. frondosa* is the only dietary source of iAs. However, it is important to emphasize that the number of samples in this study is low (*n* = 4), and the variations are large. Further, food safety issues related to sea cucumber consumption were recently reviewed by Elvevoll et al. [85] and only a few studies [17,86,87,88,89,90,91] have evaluated the content of total As in sea cucumbers, and the data on As speciation is limited [83], highlighting the need for speciation data and management, and monitoring activities of sea cucumbers.

Levels of dioxins, polychlorinated biphenyls (PCB), dioxins-like PCBs (dl-PCBs), poly aromatic hydrocarbons (PAH), per- and polyfluoroalkyl substances (PFAS) have not been published for *C. frondosa*. The EC has set a ML of dioxins and dl-PCBs of 6.5 pg toxic equivalents (TE)/g ww and 75 micrograms (µg)/g ww of ICES-6 PCB in muscle of fish and fishery products [92]. In 2018, the tolerable weekly intake (TWI) of dioxins and dl-PCBs was set (reduced from 14) to 2 pg TE per kg of body weight per week [93]. EFSA has set a new safety threshold for the four main perfluoroalkyl substances, or PFAS, that accumulate in the body. In September 2020, the EFSA re-evaluated the risk to human health from PFAS in food and established a tolerable weekly intake (TWI) of 4.4 ng per kg body weight per week. The four PFAS compounds in EFSA’s assessment are perfluorooctanoic acid (PFOA), perfluorooctane sulfonate (PFOS), perfluorononanoic acid (PFNA), perfluorohexane sulfonic acid (PFHxS) [94]. There are no ML for PAH in Echinodermata, but the EC has set a maximum level for fresh/chilled or frozen bivalve molluscs in benzo(a)pyrene (BAP) (5 µg/kg)) and the sum of benzo(a)pyrene, benz(a)anthracene, benzo(b)fluoranthene and chrysene (or PAH4) (30 µg/kg) [95]. The content of pesticides (HCH and DDT groups) is also not found specified for Echinodermata. However, for terrestrial meat a ML is noted at 0.01 mg/kg for HCH and 1 mg/kg for DDT [96,97]. Our results (Table 5 and Table 6), even if this is a single spot in time and a low number of samples, show that when compared to alternative species, POPs known to accumulate in the environment and food chain exceed the maximum level set by EU [92,95].

### 4.4. Carbon Dioxide Equivalent Footprint

The use of LCA to quantify the use of resources in production has gained widespread recognition in several sectors of industry and policy development [22,98,99]. One of the strengths of a LCA is its ability to analyse input- and output footprints in production, and thereby, as a proxy, assess the impact on the environment. A LCA can assist in several different aspects: Identify optimal opportunities to improve the environmental performance of products at various points in their life cycle; inform policy development by decision-makers in industry, government, or non-government organization; and assist in the selection of relevant indicators of environmental performance, including measurement techniques as well as marketing [22].

The estimated GHG emissions from *C. frondosa* at 7.8 kg CO_2_eq. per kg edible body-wall surpasses estimated emissions for traditional Norwegian seafood like demersal fish (1.6–2.5 kg CO_2_eq. per kg edible product) and pelagic fish (1.1–1.4 kg CO_2_eq. per kg edible product) multiple times. On the other side, products with a low edible yield, like shrimp and king crabs, are more comparable to *C. frondosa* (as long as their by-products are not utilised) [48]. Fuel for fisheries is often found to be a major emission source in seafood product value chains [100,101,102,103], as in the production of *C. frondosa*—representing 90% of total emissions. Emissions from storage (less than 0.0%), transport (1%) and cooling agents (1%) were of little influence, while processing of *C. frondosa* had some influence on total emissions (8%). Processing is an uncommon emission source from other Norwegian seafood products as the majority of seafood products are exported fresh/frozen [104,105], while *C. frondosa* is expected to be cooked and dried before export. Though, as *C. frondosa* is exported as a dried product, the natural degradation process will not be of the same concern as for fresh exported seafood [106]. *C. frondosa* might therefore be transported by more time-consuming transport methods with lower emissions, such as trains or ships [63]. Compared to Norwegian terrestrial animal protein sources, emission estimates for *C. frondosa* are similar to high estimates of pork (3–8.0 kg CO_2_eq. per kg edible product) and two to six times that of chicken (1.3–4.3 kg CO_2_eq. per kg edible product) [21,66,107]. There is a lack of post-farm data for Norwegian beef production [66]. However, compared to West-European beef production at 28.3 kg CO_2_eq. per kg edible product [21,108], the utilisation *C. frondosa* emits far less.

#### Uncertainties in Carbon Emission Footprint

Carbon emissions calculations are vastly impacted by the catching method and the efficiency achieved. As shown by the MFRI [13], the catch of *C. frondosa* may vary from 47 kg to 916 kg per hour. As fuel is the dominating emission source—this variation will severely affect the total emissions. The MFRI [13] also note how larger boats have become more prevalent in Icelandic *C. frondosa* fisheries to operate in worse weather conditions. However, this will result in higher total emissions if their catch does not increase correspondingly with fuel use [43]. A second notion is an inconsistency reported for gutting yield, ranging from 26 to 50% [14,57,58], which creates uncertainty about how much product is gained per kg catch. At the highest yield could reduce the total emissions by up to 40%. The energy use for *C. frondosa* could not be found. However, compared to more complex value chains like salmon, the emission from gutting would likely be minuscule [48]. Lastly, it should be noted that trawls and dredgers have been estimated to cause an increase in aqueous CO_2_ emissions through re-mineralisation of seabed sedimented carbon [109]. This might affect marine carbon cycling, primary productivity and biodiversity. It might also cause direct emissions into the atmosphere, although the proportion of aqueous CO_2_ that is released into the atmosphere is unknown [109]. Thus, increasing the use of demersal trawls and dredgers (for sea cucumbers) in Norwegian fisheries might result in higher emissions than assessed.

## 5. Conclusions

The aim of this article was to assess the sustainable food potential of Norwegian orange-footed sea cucumber (*Cucumaria frondosa*) for human consumption by evaluating different aspects. The body-wall of *C*. *frondosa* is a low fat and mediocre protein source with average micronutrient content. A high percentage of polyunsaturated fatty acids (PUFA) was found, though with the low total fat, the PUFA content in *C. frondosa* becomes of less interest. When graded with a nutrient density score, *C. frondosa* scores above average when nutrients are compared towards daily recommended intakes (DRI) for men and women (age 31–60) but below when the nutrient score was capped at 100% of DRI. The nutrients found to contribute most to the DRI were vitamin B12 and vitamin E (more than the DRI per 100 g body-wall), and selenium (almost 40% of the DRI per 100 g body-wall). All essential amino acids were also present in the *C. frondosa* body-wall, though their amount was a quarter of traditional animal protein sources. From a food safety perspective, all contaminants analysed were found to be in low quantity, with the exception of inorganic arsenic (iAs) as the content exceeds the limits deemed safe by food authorities. One should thus avoid consuming more than 25 g per day of *C. frondosa*. The most likely production chain for Norwegian *C. frondosa* was estimated to be a dredge fishery- and processing in Norway, and export for retail in Shanghai. This resulted in a carbon footprint several times higher than other marine protein sources (demersal and pelagic fish) and higher than terrestrial ones (pork and chicken), except for beef. Any further research should further evaluate the content of As speciation in *C. frondosa*; if there is any meaningful way of reducing the iAs content in the edible parts of *C. frondosa* through either processing or gutting; and evaluate if a low carbon footprint within the fishery stage could be achieved.

## Figures and Tables

**Figure 1 foods-12-00114-f001:**
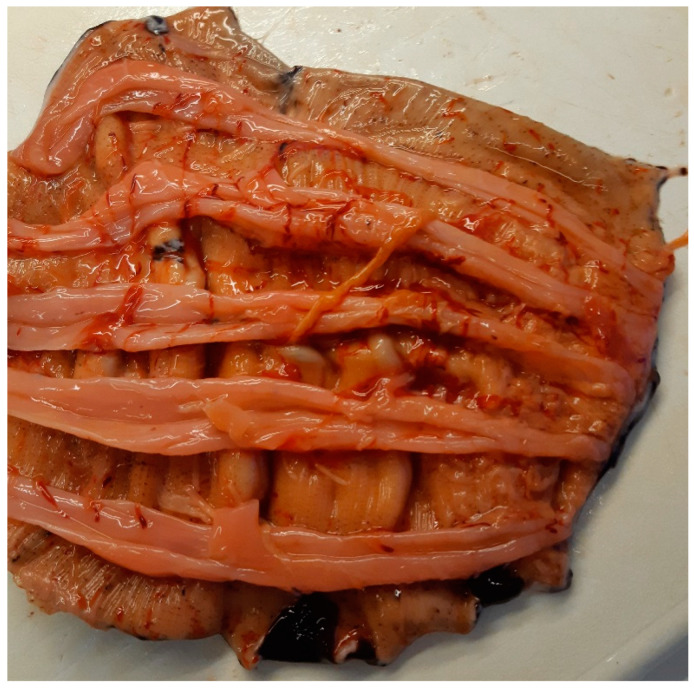
Gutted orange-footed sea cucumber (*Cucumaria frondosa*) (Photo: A. Langdal).

**Figure 2 foods-12-00114-f002:**
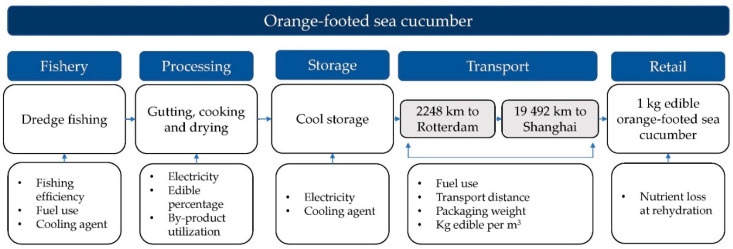
Flowchart of modelled production chain for orange-footed sea cucumber (*Cucumaria frondosa*) between cradle and retail gate (People’s Republic of China).

**Figure 3 foods-12-00114-f003:**
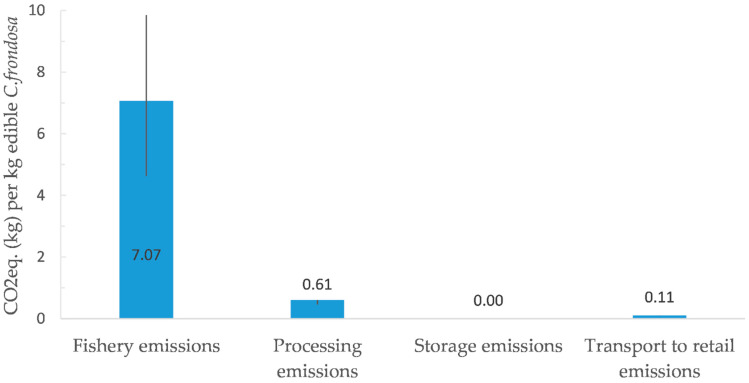
Estimated carbon dioxide equivalent (CO_2_eq.) emissions for orange-footed sea cucumber (*Cucumaria frondosa*) at the fishery-, production- (cooking and drying), storing-, and transport stage. The error bars represent the minimum and maximum estimates.

**Figure 4 foods-12-00114-f004:**
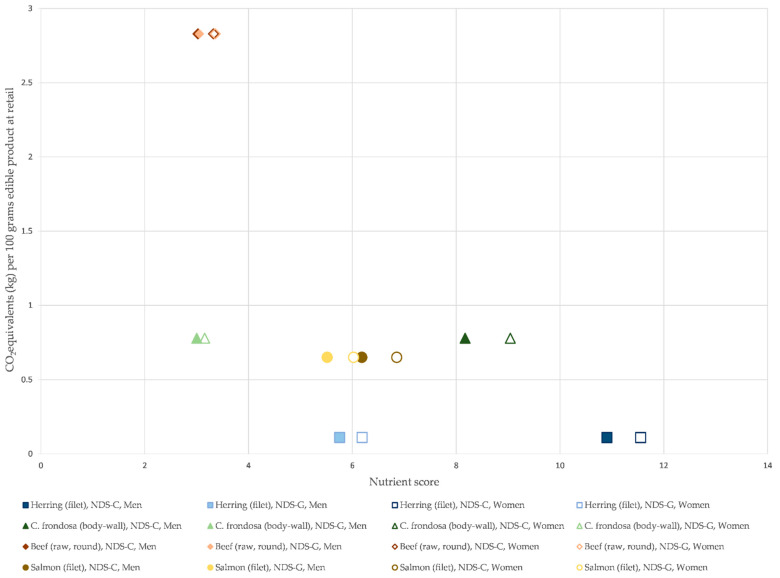
Nutrient density score (NDS) C and G, adapted from [21], for men and women aged 31–60 and compared with the carbon dioxide equivalent (CO_2_eq.) emissions between cradle (farm/fishery) and retail. CO_2_eq. emissions for herring (*Clupea harengus*) and salmon (*Salmo salar*) are adapted from [48], emissions for beef (Norwegian Red cattle (dual purpose cattle)) are adopted from [21,66], and emissions from orange-footed sea cucumber (*Cucumaria frondosa*) are from this article. NDS-C and G score food products according to their nutrient composition; however, NDS-G caps the score of nutrients which surpasses 100% of the daily recommended intake. The nutrients for herring, salmon and beef are adopted from [38] and shown in Table A2. Both the nutrient density score and the CO_2_eq. emissions are calculated per 100 g of edible product.

**Table 1 foods-12-00114-t001:** Proximate composition of lipid, protein, ash and water (g per 100 g wet weight) of orange-footed sea cucumber (*Cucumaria frondosa*) body-wall (average ± SD, *n* = 4).

Macronutrient	*C. frondosa* (Body-Wall) (g per 100 g)
Lipid	1.1 ± 0.1
Protein	6.7 ± 0.4
Ash	3.1 ± 0.0
Water	82.8 ± 0.3

**Table 2 foods-12-00114-t002:** Fatty acid composition (percentage of total fatty acids) and amount fatty acid (mg per 100 g) of wet weight orange-footed sea cucumber (*Cucumaria frondosa*) body-wall (average ± SD, *n* = 4).

Fatty Acids	*C. frondosa* (Body-Wall)	
	Amount (mg per 100 g)	Percentage of Fatty Acids (%)
Unidentified	36.4	15 ± 0.2%
C14:0	11 ± 0.6	4.4 ± 0.1%
C16:0	12.2 ± 3.6	5.1 ± 1.4%
C18:0	19.5 ± 2.3	8.0 ± 0.9%
C20:0	6.4 ± 3.3	1.3 ± 0.1%
C22:0	2.7 ± 0.2	1.1 ± 0.1%
Total SFA	48.3 ± 1.4	20.7 ± 0.5%
C16:1 n-7	12.9 ± 1.0	5.3 ± 0.3%
C18:1 n-9	5.3 ± 0.3	2.2 ± 0.1%
C18:1 n-7	6.0 ± 0.3	2.4 ± 0.1%
C20:1 n-9	2.0 ± 0.1	0.8 ± 0.0%
C22:1 n-11	2.2 ± 0.2	0.9 ± 0.0%
C22:1 n-9	3.8 ± 0.1	1.6 ± 0.1%
C24:1 n-9	4.1 ± 0.3	1.7 ± 0.1%
Total MUFA	36.3 ± 0.3	15.5 ± 0.5%
C18:2 n-6	2.1 ± 0.2	0.9 ± 0.0%
C18:4 n-3	19.2 ± 0.8	7.9 ± 0.4%
C20:2 n-6	2.0 ± 0.1	0.9 ± 0.0%
C20:4 n-6	15.6 ± 0.9	6.5 ± 1.8%
C20:5 n-3	69.8 ± 4.0	28.9 ± 1.8%
C22:6 n-3	3.4 ± 0.2	1.4 ± 0.1%
Total PUFA	112.2 ± 1.0	48.1 ± 0.5%
Total n-3	92.4 ± 5.0	38.2%
Total n-6	19.7 ± 1.1	8.3%
n.6/n-3	0.2%	

SFA indicates saturated fatty acids, MUFA indicates monosaturated fatty acids, PUFA indicates polyunsaturated fatty acids, total n-3 indicates total omega 3 fatty acids and total n-6 indicates total omega 6 fatty acids.

**Table 3 foods-12-00114-t003:** Amino acid composition (mg per g) of wet weight orange-footed sea cucumber (*Cucumaria frondosa*) body-wall (average ± SD, *n* = 4).

Amino Acids	*C. frondosa* (Body-Wall) (mg per g)
Histidine (His)	0.6 ± 0.3
Isoleucine (Ile)	2.3 ± 0.1
Leucine (Leu)	3.4 ± 0.2
Lysine (Lys)	2.7 ± 0.5
Methionine (Met)	1.0 ± 0.1
Phenylalanine (Phe)	2.1 ± 0.1
Threonine (Thr)	3.2 ± 0.1
Valin (Val)	3.2 ± 0.1
Total essential amino acids	18.5 ± 0.0
Arginine (Arg)	10.1 ± 0.9
Alanine (Ala)	3.5 ± 0.4
Aspartic acid (Asp)	5.3 ± 0.3
Cystine (Cys)	0.3 ± 0.2
Glutamic acid (Glu)	11.0 ± 0.6
Glycine (Gly)	8.3 ± 1.8
Proline (Pro)	5.9 ± 1.1
Serine (Ser)	3.9 ± 0.3
Tyrosine (Tyr)	0.4 ± 0.5
Total amino acid	67.2 ± 4.4
Total essential amino acids (mg) per g protein	275

Tryptophan is denatured during acid hydrolysis and is therefore not included in the table. Glutamine and asparagine are deaminated during acid hydrolysis and are therefore included in glutamate and asparagine acid.

**Table 4 foods-12-00114-t004:** Trace elements composition (mg per kg) of wet weight orange-footed sea cucumber (*Cucumaria frondosa*) body-wall (average ± SD, n = 4). All values below or equal to the level of detection (LOD) were set equal to LOD.

Trace Elements	*C. frondosa* (Body-Wall) (mg per kg)
Silver (Ag)	4.38 × 10^−3^ ± 3.62 × 10^−3^
Arsenic (As)	1.49 ± 0.44
(*Inorganic arsenic*) (*iAs*)	(*0.73 ± 0.48*)
Arsenic (As) *	1.77 ± 0.45
Cadmium (Cd)	0.07 ± 0.01
Cadmium (Cd) *	0.10 ± 0.01
Cobalt (Co)	0.05 ± 0.01
Copper (Cu)	0.30 ±0.02
Iron (Fe)	3.86 ± 0.33
Mercury (Hg)	1.29 × 10^−3^ ± 9.91 × 10^−5^
Mercury (Hg) *	1.3 × 10^−3^ ± 2.44 × 10^−4^
Manganese (Mn)	0.10 ± 0.01
Molybdenum (Mo)	0.07 ± 0.02
Nickel (Ni)	0.07 ± 0.02
Lead (Pb)	0.01 ± 0.00
Lead (Pb) *	0.02 ± 0.01
Selenium (Se)	1.17 ± 0.23
Selenium (Se) *	1.46 ± 0.51
Vanadium (V)	0.06 ± 0.01
Zinc (Zn)	7.90 ± 1.28

* were analysed by NILU using the methods previously described by Eilertsen et al. [33]. The rest were analysed by the Institute of Marine Research in Bergen, Norway with methods previously described by Wiech et al. [32]. Wilcoxon signed-rank test indicated no significant difference between the two analysis results on As, Cd, Hg, Pb, and Se.

**Table 5 foods-12-00114-t005:** Sum concentrations (pg toxic equivalents (TE)/g wet weight) of dioxins and furans (PCDD/PCDF), dioxin-like PCB (dl-PCB), and the total sum TE, for orange-footed sea cucumber (*Cucumaria frondosa*) (average ± SD, *n* = 4). All values below or equal to the level of detection (LOD) were set equal to LOD.

	*C. frondosa* (Body-Wall) (pg TE/g)
Sum PCDD/PCDF	0.090 ± 0.015
Sum dl-PCB	0.021 ± 0.004
Total sum TE	0.110 ± 0.015

Sum of PCDD/PCDF includes 2378-TCDD, 12378-PeCDD, 123478-HxCDD, 123678-HxCDD, 123789-HxCDD, 1234678-HpCDD and OCDD; 2378-TCDF, 12378-PeCDF, 23478-PeCDF, 123478-HxCDF, 123678-HxCDF, 123789-HxCDF, 234678-HxCDF, 1234678-HpCDF, 1234789-HpCD, OCDF. Sum dl-PCB includes PCB-77, -81, -126, -169, -105, -114, -118, -123, -156, -157, -167, and -189.

**Table 6 foods-12-00114-t006:** Concentrations (ng/g) ICES-6 PCBs, Sum PAH4, benzo(a)pyrene, Sum HCHs, sum DDT, PAH and PFAS and various pesticides for wet weight orange-footed sea cucumber (*Cucumaria frondosa*) (average ± SD, *n* = 4). All values below or equal to level of detection (LOD) were set equal to LOD.

	*C. frondosa* (Body-Wall) (ng/g)
Sum ICES-6 PCB	0.13 ± 0.06
Sum PAH4	0.68 ± 0.11
*(Benzo(a)pyrene (BaP))*	*(0.10 ± 0.03)*
Sum HCH	0.008 ± 0.001
Sum DDT	0.103 ± 0.002
All PFAS	<LOD (<0.008 to <0.03)

Sum ICES-6 PCB indicate PCB-28, -52, -101, -138, -153 and -180. Sum PAH4 indicate benzo(a)pyrene, benz(a)anthracene, benzo(b)fluoranthene and chrysene. Sum HCH indicate a-HCH, b-HCH and g-HCH. Sum DDT indicate o.p″-DDE, p,p″-DDE, o,p″-DDD, p,p″-DDD, o,p″-DDT and p,p″-DDT. PFAS content and their limit of detection (LOD) were < 0.0017 ng/g ww for 4:2 FTS, 6:2 FTS, 8:2 FTS, 10: 2 FTS, PFPS, PFHpS, PFNS, PFDS, PFHxA, PFHpA, PFDoDA; <0.008 ng/g ww for PFBS, PFHxS, PFOSlin, sum PFOS, PFOA, PFNA, PFDA, PFUnDA; and <0.03 ng/g ww for PFTrDA, PFTeDA, PFHxDA, PFOcDA, FOSA.

**Table 7 foods-12-00114-t007:** Amount of micro- and macronutrients per 100 g wet weight body-wall of orange-footed sea cucumber (*Cucumaria frondosa*). Nutrient categories were adapted from Hallström et al. [21].

Nutrients	*C. frondosa* (Body-Wall)
Qualitative Nutrients	
Protein (g)	6.7 ^2^
Fibre (g)	n/a
Omega-3 fatty acids (g)	0.09 ^2^
Retinol eq. (vitamin A) (µg)	n/a
Vitamin D (µg)	n/a
Vitamin E (mg)	28.67 ^3^
Thiamin (mg)	0.02 ^3^
Riboflavin (mg)	0.09 ^3^
Ascorbic acid (vitamin C) (mg)	n/a
Niacin equivalents (mg)	0.59 ^3^
Pyridoxine (vitamin B6) (mg)	0.099 ^3^
Cobalamin (vitamin B12) (µg)	8.6 ^3^
Folate (µg)	15 ^3^
Phosphorus (P) (mg)	38.77 ^1^
Iodine (I) (µg)	n/a
Iron (Fe) (mg)	0.37 ^1^
Calcium (Ca) (mg)	4.44 ^1^
Potassium (K) (g)	0.03 ^1^
Copper (Cu) (mg)	0.01 ^1^
Magnesium (Mg) (mg)	23.75 ^1^
Selenium (Se) (µg)	24.55 ^1^
Zinc (Zn) (mg)	0.57 ^1^
Dis-Qualitative Nutrients	
Saturated fatty acids (g)	0.05 ^2^
Sodium (g)	n/a

^1^ adopted from Song et al. [17] and converted to wet weight by expecting 82.8% water in fresh product, ^2^ analysed at UiT—The Arctic University of Tromsø, and ^3^ analysed by an accredited laboratory at the Institute of marine research in Bergen. Cells noted n/a represents data that were not accessible.

## Data Availability

The data presented in this study are openly available in DataverseNO at doi.org/10.18710/D4GLZZ.

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
