# Peer review of "Climate Performance, Environmental Toxins and Nutrient Density of the Underutilized Norwegian Orange-Footed Sea Cucumber (Cucumaria frondosa)"

_foods, 2022, doi:10.3390/foods12010114_

Round 1

Reviewer 1 Report

The work seems very interesting but the way it is written denotes a lot of fragility. The topic of materials and methods should be reformulated, they should be described in such a way that it is possible to reproduce them in a simple and clear way. Furthermore, the writing of the document should be revised, it should be more careful and scientific. Only after a thorough reformulation I will proceed with the careful evaluation of the discussion of results.

Climate Performance, Environmental Toxins and Nutrient Den- 2 sity of the Underutilized Norwegian Orange-footed Sea Cu- 3 cumber (Cucumaria frondosa)

Line 61 – You should clarify the definition of heavy metals.

Line 77, 76, 78 – The first time a species is described it must be in full.

The topic of materials and methods should be reformulated, they should be described in such a way that it is possible to reproduce them, in a simple and clear way. Furthermore, the writing of the document should be revised, it should be more careful and scientific.

Line 81 - in the next paragraph it looks like a topic of "Sampling and pre-treatment" and then you should follow the topics related to "Proximity analysis" or "Nutritional composition analysis" including: 2.2.1 | Moisture content and ash content; 2.2.2 | Total fat content and fatty acid profile; 2.2.3 | Crude protein content and amino acid profile

Line 86 – Stored in what conditions?

Line 90 – What are the temperature conditions of the cycle described?

Line94-96 – Although they indicate the source of the method, it should be briefly presented and they should also indicate how the results will be expressed (in which unit).

Line 97-107 – In addition to considering that the methods are described very superficially, they do not present the units in which they are expressed nor is it understood how they are calculated.

 Line101/102 – “ a higher sample to..” – this is an identification of a very vague and dubious description.

Line 108-124 – The quantification procedure for amino acids is described, but that for total protein is not in the materials and methods. They should also indicate in which units they will be expressed. They should pay attention to the way they present the units usually hours=h; seconds =s; minuts= min;

Line 125-130 - It doesn't seem to make much sense to present the results in this way. Samples with different origins will have different nutrient contents, the same is expected when they are determined by different methods. These methods are not even clearly identified. This data presentation (table 7) is not very scientific for a research article. Furthermore, I cannot understand what the authors mean by micronutrients.Line 134 – What do you mean by “with no weighing (increasing the value of particular nutrients according 134 to the need of a population).”?

Line 139 – MR is not in the indicated table

Line 140 – I would indicate "Nutrients-scores (theoretical value)".

Line 138- What are “the dis-qualifying nutrientes”?

Line 149-154 – In addition to the fact that these elements are also anthropogenic pollutants and that a distinction needs to be made between trace elements and heavy metals, the methods must be clearly described and the units in which they are represented must be indicated. I would like to understand why in the introduction they mention Hg and here they highlight Se. They should bear in mind that all the elements determined should be identified in the methods.

Line 155-163 – Very incomplete method. What is the equipment, specific column and specifics of the method used? Units in which the results will be expressed.

Line164 -167 – If in fact the largest consumption of this species in your country comes from this place, that is correct. Otherwise it is not the right way.

Line 176 – the units are universal, it is not necessary to present them in this way, just kg

Line 164-257 - review all these methods. Units are not written out in full. Pay attention to how molecular formulas are written throughout the document.

Line 261 – Amend grams to g, when g refers to centrifugation it should appear in italics.

Line 262 - Figure 3 - there is no point in presenting this graph, the information in the text is sufficient.

Line 267 – What does the remaining 94% represent?

Line 279 – presentation of units

In table 2 it indicates that the results are by body-wall, but in the methods it indicates that it is by fillets or edible part, please be consistent. The same happens in other results.

line 283 – what do you mean by “consisting of 19 mg of 283 essential amino acids.”

Line 285 –  I suggest changing to “ The essential aminoacid identified to total amino acids ratio…”

Line287 - I suggest changing to “Table 3. Amino acid composition (mg / g of wet weight) orange-footed sea cucumber…”

Table 3 - I don't understand what the division between amino acids is due to and in the final line milligrams must have lowercase m.

Table 4 – the “<0.00±0.00” represents values below the detection limit? If yes, it must be identified. If no, it must show the value in exponent.

Table 7 – iodine i tis na importante elemet in marine resources. What represents n/a?

Figure 5 - please replace Kg with kg

Reviewer 2 Report

The authors of Climate Performance, Environmental Toxins and Nutrient Density of the Underutilized Norwegian Orange-footed Sea Cucumber (Cucumaria frondosa) are assessing the carbon footprint of harvesting of orange-footed sea cucumber together with analysis of macro- and micronutrients as well as heavy metals. It was interesting to see the assessment/ calculation of carbon footprint for this commodity; however, the manuscript lacks an overall aim and conclusion. The authors report various results but don’t provide comparison with already published data, so it is difficult to see where studied sea cucumbers stand. Also, if one of the aims was to explore sea cucumbers nutrient potential to become a food source on a major scale, the authors should provide some summary statement if the presented data support this aim. It seems like a great source of Vit B12 and E; however, the suggested portion of 100 g would cause high load of As as it was concluded that no more than 25 g per 60 kg of body weight should be consumed per day. It seems that without having a clear aim of the study, the manuscript is lacking focus and it difficult to figure out what is the message authors are trying to put across. I also found the Materials and method section very poorly written. I understand that the authors are using previously published methods, but that doesn’t justify not including description of sample preparation and analysis in the body of the manuscript. Putting a reference of the publication from which the method was used without a brief description of what was done is unacceptable. As this section is written right now, it makes an impression that not quality control was performed during the analysis so please add this section into the manuscript. Clarification is also needed in the use of n=4. Are the author using n as number of replicates or as number of samples analysied. In the sample collection, it was stated that 61 sea cucumbers were collected, and the weight was measured in 61 of them: however, all other analysis states n = 4. Does it mean that only 4 samples from 61 were subjected to the given analysis?

In general, there are inconsistencies in the use of tense through the manuscript, interchanging between present and past tense. A thorough revision of the tense used is needed and as it is a convention, the manuscript should be written in past tense.

Please see some specific comments for individual sections and more comments within the actual body of the manuscript.

Abstract: I would recommend to re-write an abstract for this manuscript and closely follow the guidelines of what should be included in the abstract. The authors presented very little of actual data in the abstract what leaves readers with very limited knowledge of what was achieved by the research.

Introduction: Usually, the introduction ends with a statement of what the aim of the presented study was and what contribution this study may have to the society/ general public/ scientific community some sort of reasoning why this study was important to carry out. Because the introduction is lacking such a statement, it is difficult to assess whether the presented results are relevant to the aim of the study. I would suggest adding this paragraph, so it gives a clear focus to the manuscript.

Materials and methods: please give a brief description of methods which were used not only a reference to the source. It is impossible to assess the credibility of results if the analytical methods for sample preparation and measurement are not described.

Results: Figure 1.: figure is missing axis and axis’ titles should be corrected or added. Please see the manuscript for more comments.

Table 4. Without having details on sample preparation and analysis is it not possible to assess the feasibility of the data. Please, remove SD from values which are less than 0.00. Due to formatting of the values to 2 decimal places some of the SD are 0.00. I would suggest formatting the number to significant figures rather than decimal places.

Discussion: There are some inconsistences within the text and lack of comparison with published data. The reader doesn’t know where the presented data stand among what was already published. Some previous work is being cited but there is no comparison presented. It is not clear why the authors analysed only 4 samples for iAs and other metals if 61 sea cucumbers were collected. In my personal view, drawing any meaningful conclusion on the analysis of 4 samples is difficult to justify.

Conclusion: because there was no aim of the work clearly stated, it is difficult to assess what the conclusion is. Also, the conclusion is lacking future ideas for research, suggestions for some improvements or recommendations.

Reviewer 3 Report

Reviewer Comment

Journal: Foods

Dear Editor,

This paper deals the Climate Performance, Environmental Toxins and Nutrient Den- 2 sity of the Underutilized Norwegian Orange-footed Sea Cu- 3 cumber (Cucumaria frondosa)by Andreas Langdal et al.

The following is a report on the above manuscript, I recommend reconsideration of the manuscript after minor Revision.

Points to be corrected:

1- The scientific content of the manuscript is good.

2- The abstract does not report the most important results of the work

3-the novelty of the work has not been demonstrated appropriately in the introduction section.

4- RESULTS and DISCUSSION section. The authors used and cite references, but it is not enough for suitable scientific discussion. The authors have to cite more related and relevant papers and give more discussion and evidence for their claims.

6-What is the number of replicates for each test?

7- The author should present the figures with high resolution

Round 2

Reviewer 2 Report

The revised manuscript is significantly improved however there are still several issues that need to be addressed.

1.)      Use of present of future tense should be avoided and past tense should be used

2.)      ICP-MS analysis requires more details

a.       ICP-MS detection modes – please provide details if collision or reaction gases were used if so which one for which elements and at what setting

b.      Were any certified reference materials used during the sample preparation, if yes, please provide details and if not, please justify the accuracy of the reported results and what kind of validation was done

c.       How was the LOQ calculated, what quantitation method was used, were any method blanks included in the sample prep if so what are the results

3.)      The authors are not consistent with use of abbreviations – arsenic is abbreviated in line 62 but used without abbreviation in the rest of the manuscript. Please review the entire manuscript and change these inconsistencies. Please note that this is not only for arsenic but all of the abbreviations.

4.)      Please unify the use of sea cucumber/ Cucumaria frondosa/ C. frondosa – the species of sea cucumber studied was defined in the introduction and method section so though the text only one of these names should be used.

5.)      Line 581: please fix the font color and size

More comments can be found in the pdf version
